# SMOOTHING AND SPATIAL VERIFICATION OF GLOBAL FIELDS

#### **Gregor Skok**

University of Ljubljana, Faculty of Mathematics and Physics Jadranska Cesta 19, 1000 Ljubljana, Slovenia Gregor.Skok@fmf.uni-lj.si

#### Katarina Kosovelj

University of Ljubljana, Faculty of Mathematics and Physics Jadranska Cesta 19, 1000 Ljubljana, Slovenia Katarina.Kosovelj@fmf.uni-lj.si

Corresponding author: Gregor Skok, Gregor.Skok@fmf.uni-lj.si.

#### ABSTRACT

Forecast verification plays a crucial role in the development cycle of operational numerical weather prediction models. At the same time, verification remains a challenge as the traditionally used nonspatial forecast quality metrics exhibit certain drawbacks, with new spatial metrics being developed to address these problems. Some of these new metrics are based on smoothing, with one example being the widely used Fraction Skill Score (FSS) and its many derivatives. However, while the FSS has been used by many researchers in limited area domains, there are no examples of it being used in a global domain yet. The issue is due to the increased computational complexity of smoothing in a global domain, with its inherent spherical geometry and non-equidistant and/or irregular grids. At the same time, there clearly exists a need for spatial metrics that could be used in the global domain as the operational global models continue to be developed and improved, along with the new machine-learning-based models. Here, we present two new methodologies for smoothing in a global domain that are potentially fast enough to make the smoothing of high-resolution global fields feasible. Both approaches also consider the variability of grid point area sizes and can handle missing data appropriately. This, in turn, makes the calculation of smoothing-based metrics, such as FSS and its derivatives, in a global domain possible, which we demonstrate by evaluating the performance of operational high-resolution global precipitation forecasts provided by the European Centre for Medium-Range Weather Forecasts.

**Keywords** smoothing · global domain · forecast verification · global forecasting · spatial verification

#### 1 Introduction 19

10

12

15

- Forecast verification plays a crucial role in the development cycle of operational numerical weather prediction models. 20
- At the same time, verification remains a challenge as the traditionally used non-spatial forecast quality metrics, such as 21
- the Root-Mean-Square-Error metric [RMSE, Wilks, 2019], that only compare the values of the observed and forecasted 22
- fields at collocated locations, exhibit certain drawbacks. One example is the so-called 'double penalty' issue, which 23
- penalizes forecasts for both false alarms and missed events. Another is the difficulty distinguishing between near misses 24
- and substantial spatial displacements [Brown et al., 2012, Skok, 2022]. 25
- This is why different spatial verification measures have been developed over the years. These try to address the problems
- of the non-spatial metrics by comparing not only the values at collocated locations but also taking into account values 27
- at other locations. Depending on how they work, they can be classified into five categories [Gilleland et al., 2009, 28 29 Dorninger et al., 2018]: scale separation/decomposition metrics [e.g., Casati et al., 2004, Mittermaier, 2006, Casati,
- 2010, Buschow and Friederichs, 2021, Casati et al., 2023], feature-based approaches [e.g., Ebert and McBride, 2000, 30

- Davis et al., 2006a,b, Wernli et al., 2008, Davis et al., 2009, Wernli et al., 2009], field deformation techniques [e.g.,
- Keil and Craig, 2007, 2009, Marzban et al., 2009] and distance metrics [e.g., Baddeley, 1992, Gilleland, 2017] and the
- neighborhood methods [e.g., Roberts and Lean, 2008, Roberts, 2008, Skok, 2022].
- To our knowledge, examples of high-resolution global fields analyzed by spatial metrics that adequately account for
- Earth's spherical geometry are almost non-existent in the published literature (except for Skok and Lledó [2025] and
- possibly Mittermaier et al. [2016]). We have identified two likely reasons for this gap: existing methods are designed for
- planar geometry, and adapting them to the non-planar geometry of a global domain is challenging, and the computational
- complexity in spherical geometry significantly increases, rendering the use with contemporary state-of-the-art global
- high-resolution models prohibitively expensive [Skok and Lledó, 2025]. At the same time, there clearly exists a need
- for spatial metrics that could be used in the global domain as the operational global models continue to be developed
- and improved along with the new machine-learning-based models [e.g., Weyn et al., 2020, Bi et al., 2023, Lam et al.,
- 2023, Lang et al., 2024] that also show increasing potential for global forecasting [Skok and Lledó, 2025].
- The Fraction Skill Score [FSS, Roberts and Lean, 2008, Roberts, 2008] is a widely used neighborhood-based verification
- metric. It works by first applying a threshold, thereby converting the original fields to binary fields, and then calculating
- the fractions that represent the ratio between the number of non-zero and all points located inside a neighborhood of
- prescribed shape and size, which are then used to calculate the score's value. We note that calculating the fraction
- values from a binary field is mathematically equivalent to smoothing the binary field using a constant value smoothing
- kernel of the same shape and size as the neighborhood. FSS is a popular metric with many derivatives, as different
- researchers have tried to extend its functionality by developing new scores based on the same fundamental principles,
- for example, to extend the original FSS to be able to analyze ensemble/probabilistic forecasts [e.g., Zacharov and
- Rezacova, 2009, Schwartz et al., 2010, Duc et al., 2013, Bouallègue et al., 2013, Dey et al., 2014, 2016, Ma et al., 2018,
- Gainford et al., 2024, Necker et al., 2024], to verify non-scalar variables [e.g., wind, Skok and Hladnik, 2018], to
- also take into account timing errors [e.g., Duc et al., 2013, Ma et al., 2018, Mittermaier, 2025], to provide an estimate
- of forecast displacement [e.g., Skok and Roberts, 2018, Skok, 2022], to provide localized information on forecast
- quality [Woodhams et al., 2018, Gainford et al., 2024, Mittermaier, 2025], or to develop other similar smoothing-based
- metrics with somewhat different requirements and properties [e.g., ones that do not necessarily require thresholding, for
- example, Skok, 2022].
- Conceptually, employing the FSS or one of its derivatives in spherical geometry poses no inherent issues; however,
- challenges emerge due to the increased computational complexity of smoothing (fraction calculation), which is
- computationally the most expensive part of the score's calculation. Namely, for a regular and equidistant grid, the
- smoothing can be done very efficiently using either the summed-fields approach [Faggian et al., 2015] with time
- complexity O(n) or by using the Fast-Fourier-Transform-based convolution [Smith, 1999] with time complexity
- $O(n \log(n))$ , with n being the number of points in a field. The problem is that these approaches cannot be used on
- a sphere because the grid is inherently non-equidistant and/or irregular. Using the so-called explicit summation for
- smoothing (where at each location, the distance to all other grid points is calculated to determine which fall inside
- the smoothing kernel) is still possible, but becomes prohibitively expensive for global high-resolution-model fields
- consisting of millions of points due to its time complexity of  $O(n^2)$ .
- An additional complication in spherical geometry is the variability of grid point area sizes. Namely, in a global domain,
- the area size represented by each grid point is usually not the same for all grid points. If the smoothing is done in a way
- that does not account for this, the spatial integral of the field could change considerably as a result of the smoothing.
- For example, smoothing a precipitation field could cause the total volume of precipitation in the domain to increase or
- decrease. To alleviate this issue, the smoothing method needs to be area-size-informed.
- This paper aims to develop novel computationally efficient methodologies for smoothing fields on a sphere. Such
- methodologies are required for the smoothing-based verification metrics, such as FSS and its many derivatives, to be
- used to evaluate the forecast performance of state-of-the-art operational global high-resolution models. The smoothing
- methodologies must also be area-size-informed and preferably able to handle missing data values appropriately.

### 2 Area-size-informed smoothing

The area-size-informed smoothed value at grid point i can be calculated as

$$f_i'(R) = \frac{\sum\limits_{j \in \mathbb{K}_i(R)} f_j a_j}{\sum\limits_{j \in \mathbb{K}_i(R)} a_j},\tag{1}$$

where  $f_j$  is the field value at point j,  $a_j$  the area size representative for point j, and  $\mathbb{K}_i(R)$  the subset of all points around point i, for which the great circle distance (along the spherically curved surface of the planet) to point i is

Figure 1: A visualization showcasing the area-size-informed smoothing methodology in two-dimensions. The small circles denote the grid points, while the polygons represent the corresponding Voronoi cells (defined as the region that is closest to the corresponding grid point). The large circle represents the smoothing kernel around the point denoted by a + sign, while the Voronoi cells of points inside the kernel are colored in gray.

less than R. In other words, the smoothed value represents the area-size-weighted average value of points inside a

spherical-cap-shaped smoothing kernel centered on the selected point. The radius of the smoothing kernel can also be

called a smoothing radius.

Fig. 1 showcases an example of area-size-informed smoothing in the case of an irregular grid in two dimensions. Since

the grid is irregular, the area sizes of points, denoted by the corresponding Voronoi cells, differ. In this case, the subset 85

of points inside the smoothing kernel, denoted as  $\mathbb{K}_i(R)$  in Eq. 1, is shown by the gray color, while the rest of the points 86

are white. 87

Fig. 2 shows some examples of smoothed fields of forecasted 6-hourly accumulations of precipitation produced by

the high-resolution deterministic Integrated Forecasting System [IFS, ECMWF, 2023a,b] of the European Centre

for Medium-Range Weather Forecasts (ECMWF). The IFS is considered one of the best-performing operational

medium-range global deterministic models and is frequently used as a benchmark to which other models are compared

against [e.g., Bi et al., 2023, Lam et al., 2023, Lang et al., 2024], which makes it especially suitable to be used as an

example.

The IFS uses an octahedral reduced Gaussian grid O1280 [Malardel et al., 2016], which consists of around 6.5 million 94

grid points. The points are arranged in fixed-latitude circular bands, with the band closest to the equator consisting of

5136 equidistant points spread around the Earth. In the poleward direction, each next band has four points fewer than

the previous one, with the last band, located close to the poles, consisting of only 20 points. This setup makes the grid

irregular, with area size of the points also varying substantially with latitude, from 61 km<sup>2</sup> at the equator, to 93 km<sup>2</sup> 98

at 75°, where it is the largest, to 18 km<sup>2</sup> close to the poles, where it is the smallest [Skok and Lledó, 2025]. The IFS 99 100

precipitation data was provided to us by the ECMWF in the form of netCDF files that contained the lat-lon locations

of the points, the precipitation accumulation values, and the area size data of all the points. All the numeric data was

provided in float32 numeric format.

The smoothing methodology represented by Eq. 1 does not have any limitations or requirements about the grid being

regular - the only assumption is that the points are located on a sphere (in our case, we also assumed that the sphere

radius was equal to the Earth's radius). It is worth noting that the smoothing methodology does not require the

connectivity information. The only data required for calculating the smoothed values are the original field values, the 106

locations of all the points on the sphere, and their area size information. In the case of IFS fields, the area size data was 108 already provided by the ECMWF, but if it was not available, it could be obtained by performing the Voronoi tessellation

on the sphere, for example. 109

Our computational setup consisted of a computer with an AMD Ryzen Threadripper PRO 5975WX processor with 32 110

physical cores. The Debian 12 Linux operating system was installed on the computer. The code was written in C++,

and the gcc compiler version 12.2 was used to compile the code with the OpenMP programming interface used for

shared-memory multi-thread computing. Hyper-threading was enabled. Even though the IFS data was provided in

float32 format, we consistently used double (float64) precision in the C++ code, except in one special case (for more

information, please refer to the "Code and data availability" section). 115

Due to the spherical periodicity of the global domain, the smoothing kernel with  $R \ge 20\,000$  km will cover the whole 116

surface of the Earth (i.e., in this case,  $\mathbb{K}_i(R)$  is guaranteed to contain all the grid points), resulting in the smoothed 117

value being the same everywhere - the so-called asymptotic smoothing value, which we denote as  $f'_{asy}$ . The asymptotic value can be calculated easily with time complexity O(n) as  $f'_{asy} = \sum f_j a_j / \sum a_j$ , where both sums go over all the 119

Figure 2: Visualization of smoothed fields of forecasts of 6-hourly precipitation accumulations in the period 00-06 UTC for 11 October 2022 by the IFS model (the forecast was initialized at 00 UTC on the same day). (a) the original non-smoothed field, (b-i) the smoothed fields using a smoothing kernel radius (R) ranging from 20 to 10 000 km. The green circle indicates the size of the smoothing kernel.

points, and  $\sum a_i$  represents the surface area of the whole Earth. Thus, as the smoothing kernel becomes larger, the field will become less variable, with the smoothed values being ever closer to the asymptotic value. For example, Fig. 2i

shows an example with the smoothing kernel radius 10000 km, which covers about half the Earth's surface, with the 122

variability of the smoothed value being very low.

120

129

135

To calculate the smoothed value via Eq. 1, the two sums over the points inside the smoothing kernel must be performed. 124

The so-called linear search approach is the most straightforward way to identify these points. In this case, a test is 126

performed for each point in the domain by calculating its distance from the point at the center of the smoothing kernel

and comparing it to the size of the smoothing kernel radius, thereby identifying the ones that satisfy this criterion.

Under the assumption that the Earth is spherical, the Great Circle Distance (GCD) between the two points can be

calculated using the latitude/longitude coordinates of both points by utilizing the Haversine formula [Markou and

130 Kassomenos, 2010]. However, using this approach, which requires the evaluation of multiple trigonometric expressions,

turns out to be computationally slow.

Alternatively, the grid points can be projected from the model's native two-dimensional spherical coordinate system 132

into a three-dimensional Euclidean space, where all the grid points are located on the surface of a sphere. In this new 133

coordinate system, the Euclidean distance between two points on the Earth's surface is the so-called tunnel distance

(TD), representing a straight line between the two points that goes through the sphere's interior. The GCD can be easily

converted to the TD or vice versa, using the relation 136

$$TD = 2r_E \sin\left(\frac{GCD}{2r_E}\right) \tag{2}$$

- or its inverse, where  $r_E$  is the Earth's radius. Since a larger GCD will always correspond to a larger TD and vice versa,
- searching for the points inside a specified search radius defined by TD in the three-dimensional space will yield the same
- results as using the corresponding value of GCD, utilizing the Haversine formula in the model's native two-dimensional
- spherical coordinate system.
- Thus, for a specific GCD value, the corresponding value of TD can be obtained via Equation 2, and used as a
- search radius in the three-dimensional Euclidean space. The square of the distance between the two points in a
- three-dimensional Euclidean space is defined as  $d^2(i,j) = (x_i x_j)^2 + (y_i y_j)^2 + (z_i z_j)^2$ , and its calculation
- does not require the costly evaluation of trigonometric functions. Moreover, the square of the distance can be directly
- compared with the precalculated square of the TD-defined search radius, thus avoiding the costly square root operation.
- This is why searching for the points inside the search radius in the three-dimensional Euclidean space is markedly faster
- than the Haversine-formula-based approach (in our case, testing showed it was approximately 50 times faster).
- Nevertheless, even with the projection into the three-dimensional Euclidean space, the smoothing via the linear search
- approach is slow. Namely, if the number of grid points in the field is n, and at each point, the distance to all the other
- points needs to be calculated, the time complexity is  $O(n^2)$ . This makes the linear-search-based approach prohibitively
- expensive for use with current state-of-the-art operational high-resolution models, which typically use grids with
- millions of points.
- For example, smoothing a precipitation field from the IFS model using a 1000 km smoothing kernel radius (Fig. 2g)
- takes about 11 hours on our computer when utilizing a single thread. The approach can be relatively effectively
- parallelized using multiple threads to parallelize the loop over all the points. Thus, using ten threads instead of just one
- reduced the computation time from 11 hours to about 1.2 hours. However, even with the parallelization, the approach is
- still too slow for operational use in a typical verification setting, as the model's performance is usually evaluated over a
- large set of cases represented by a sequence of fields from a longer time period or a wide array of weather situations.
- Thus, a clear need exists for smoothing approaches that are considerably faster.

# 160 3 K-d-tree-based smoothing

- This approach requires the points first to be projected to the three-dimensional Euclidean space in the same manner as
- described for the linear-search-based approach. Same as before, the search radius in terms of TD can be calculated
- from the GCD-defined smoothing kernel radius using Equation 2 and then used for the search in the three-dimensional
- Euclidean space.
- Identification of points that lie inside the search radius can be sped up considerably by the use of a k-d tree [short
- for a k-dimensional tree, Bentley, 1975, Friedman et al., 1977, Bentley, 1979]. A k-d tree is a multidimensional
- binary search tree constructed for each input field by iteratively bisecting the search space into two sub-regions, each
- containing about half of the nonzero points of the parent region [Skok, 2023].
- The so-called balanced k-d tree is constructed by first performing a partial sort of all the points according to the value of
- the first coordinate and then selecting the point in the middle for the first node (also called the root node), which splits
- the tree into two branches, each containing about half the remaining points. For each branch, the process is repeated by
- partially sorting the points by the second coordinate and selecting the middle point as the node again, which splits the
- remaining points into two sub-branches. The process is then repeated for the third coordinate, then again for the first
- coordinate (in case the space is three-dimensional), and so on until all the points have been assigned to the k-d tree as
- nodes.
- The time complexity of a balanced tree construction is  $O(n \log(n))$  [Friedman et al., 1977, Brown, 2015]. For example,
- constructing a balanced k-d tree for about 6.5 million points of the IFS model grid took about 2.5 seconds. Note that if
- multiple fields that use the same grid need to be smoothed, the tree can be constructed only once and kept in memory or
- saved to a disk to be reused later. Once it is needed again, it can be simply loaded from the disk, which is an operation
- with time complexity O(n).
- Once the tree is constructed, the identification of points that lie inside a prescribed search radius can be performed by
- traversing the tree starting from the root node and moving outwards by evaluating a query at each split and backtracking
- to check the neighboring branches if necessary. The search can be done in  $O(\log(n) + k)$  expected time, where k is the
- typical number of points in the search region [Bentley, 1979], as opposed to O(n) for the linear-search-based approach.
- For all but the smallest smoothing kernels  $k \gg \log(n)$ , thus the time complexity can be approximated as O(k). Since
- producing the smoothed field requires the search to be performed for all points, the expected time complexity of the
- smoothing using the k-d-tree-based approach is O(nk) as opposed to  $O(n^2)$  for the linear-search-based approach. This

- means that, for small smoothing kernels, the k-d-tree-based approach will be much faster, but for large kernels, when k becomes comparable to n, the benefit will vanish.
- Fortunately, the calculation speed can be improved further by embedding the so-called Bounding Box (BB) information
- on each tree node. The BB information consists of the maximum and minimum values of the coordinates of all the
- points on all sub-branches of a node. This information defines the extent of a multidimensional rectangular bounding
- box that is guaranteed to contain all the points in a specific branch. Adding BB information to the tree is trivial and very
- cheap since a single iterative loop over all the tree nodes is required to determine and add this data the time complexity
- of this is O(n), and thus the cost is almost negligible.
- Once the BB data is available, it can be utilized to skip the branches that are guaranteed to fall completely outside the
- sphere defined by the search radius. This can be done by first determining which corner of the BB is the closest to the
- center of the search radius sphere. Next, if the distance of this corner to the center of the search radius sphere is larger
- than the search radius, then all the points in the node's sub-branches are guaranteed to be located outside the sphere,
- meaning this branch can be ignored entirely, thus reducing the computational load.
- The BB information can also be used to identify the branches that are guaranteed to be fully inside the search radius
- sphere. This can be done by first determining which corner of the BB is the furthest away from the center of the search
- sphere. Next, if the distance of this corner to the center of the search sphere is smaller than the search radius, all the
- points in the node's sub-branches are guaranteed to be inside the sphere. This means that all the points in this branch
- can simply be added to the list of points known to be inside the search sphere without the need to do any more checks
- and distance evaluations, thus reducing the computational load.
- However, although the above-mentioned BB-information-based improvements do make the search markedly faster, the
- time complexity of the smoothing approach remains O(nk), as in the end, the sums in Eq.1 still need to be performed
- over all the points inside the search radius sphere.
- Crucially, the speed of the k-d-tree-based smoothing can be further improved by realizing that, besides the BB
- information, additional data relevant to the smoothing can be embedded into the tree. Namely, one can precalculate the
- partial sums of  $f_i a_i$  and  $a_i$  terms (from Equation 1) of all the points in the node's sub-branches and add this data to
- each node. Similarly to adding BB information to the tree, adding the partial sums data is very cheap as it requires a
- single iterative loop over all the tree nodes (the time complexity is again only O(n)).
- For branches that are fully located inside the search sphere (as mentioned above, this can be determined using the BB
- data), the partial sum information of a node can be used to account for all points in the whole branch without the need
- to dive deeper into it. Such branches, which happen to be located near the middle of the search sphere, far away from
- its border, can contain a large number of points. Thus, the reduction of computational cost can be potentially large, as
- one node can provide the sum information for many points.
- This is not true for branches with points near the border region of the search sphere, as there the algorithm needs to dive
- very deep into the tree to accurately determine which points lie inside or outside of the search region. Thus, the main
- part of the remaining computation cost can be attributed to the evaluation of points located near the search sphere's
- border region. Since the number of points in the border regions is roughly proportional to  $\sqrt{k}$ , the time complexity of
- the smoothing reduces to approximately  $O(n\sqrt{k})$ , which is a huge improvement over O(nk). If the spatial density of
- points is roughly constant,  $\sqrt{k}$  is approximately proportional to R, with R being the smoothing kernel radius, and the
- time complexity is approximately O(nR).
- For example, as already mentioned, the linear-search-based smoothing of the IFS precipitation field shown in Fig. 2a,
- for a 1000 km smoothing kernel radius, takes about 11 hours using a single thread, with the calculation time being
- similar also for other kernel sizes. In comparison, the k-d-tree-based approach takes only eight minutes using a single
- thread and about one minute if ten threads are used in parallel. As expected, the smoothing calculation is faster if a
- and about one initiate it the timeass at used in parameter. As expected, the smoothing caretration is laster if a
- smaller smoothing kernel is used. For example, using  $R=100\,\mathrm{km}$ , the calculation takes 34 s using a single thread,
- which reduces to 4.5 s if ten threads are used in parallel. On the other hand, for very large smoothing kernels, the
- k-d-tree-based approach is still markedly faster than the linear-search-based approach, but the difference is not as large
- as for the smaller kernels. For example, for a kernel with  $R = 10\,000$  km, the k-d-tree-based calculation took about 70
- minutes using a single thread and about 12 minutes if ten threads were used in parallel.
- In the end, we would like to note that we are not the first to use the k-d trees with FSS-based verification. Namely,
- Mittermaier [2025] already used k-d-trees for calculating FSS-based metrics. However, in their study, they focused
- on a limited area domain over the Maritime continent while seemingly assuming a planar geometry without properly
- taking into account the spherical geometry of the Earth. Contrary to our work, they also did not seem to actively focus
- on trying to come up with ways to make the smoothing calculation substantially faster and only used relatively small
- smoothing kernels with radii below 100 km, while at the same time relying on precalculated lookup tables of points

Figure 3: Same as Fig.1, but also showing the smoothing kernel for a second point located right of the original point (the points are marked with + signs and 1 and 2). The Voronoi cells of points located inside both kernels are shown with a light shade of gray. The darker shades of gray indicate the points inside only one kernel.

- inside the smoothing kernel, to make the smoothing calculation somewhat faster. This means the time complexity
- of their approach was limited to O(nk), which is unfortunately too slow for use with global high-resolution fields;
- moreover, for larger neighborhoods, the precalculated lookup tables would be very large and require too much memory
- in order to be used effectively.

# 246 4 Overlap-detection-based smoothing

- While the k-d-tree-based smoothing is markedly faster than the linear-search-based approach and makes the smoothing
- of high-resolution fields potentially feasible, it is still relatively slow for very large smoothing kernels, which can be
- problematic if many fields need to be smoothed. Thus, it makes sense to try to come up with a different approach that
- would be even faster.
- The alternative approach is based on identifying and then using the information on the overlap of the smoothing kernels
- centered at nearby points to increase the speed of the smoothing calculation. Fig. 3 is similar to Fig. 1 but also shows
- the smoothing kernel for a second point located to the right of the original point. The Voronoi cells with points inside
- the two kernels are colored with different shades of gray, according to the point being located inside both kernels or
- only one.
- Let us assume that for the first point, the values of two sums from Eq. 1 (i.e.,  $\sum f_j a_j$  and  $\sum a_j$ ) are known. The
- equivalent sums for the second point can be obtained by subtracting the  $f_k a_k$  or  $\overline{a_k}$  terms corresponding to the points
- that are located in the smoothing kernel of the first point but not the second (indicated by the dark gray shading on the
- left side in Fig. 3), and adding the terms corresponding to the points that are located in the kernel of the second point
- but not the first (indicated by the dark gray shading on the right side in Fig. 3). This can then be repeated for the next
- neighboring point, and so on.
- This means that the total sums (over all the points inside the smoothing kernel) must be calculated only for the first
- point (which can be randomly chosen). For all the next points, the values of the sums can be obtained with the help of
- the nearby points for which the values of the sums are already known by subtracting and adding the appropriate terms
- with respect to the overlap of the two smoothing kernels. If the two points are neighbors, the number of terms that need
- to be subtracted and added can be approximated by the number of points that comprise the border of the smoothing
- kernel area, which is approximately proportional to  $\sqrt{k}$ .
- Evaluating the overlap of the smoothing kernels of nearby points and determining which terms need to be subtracted
- or added can be done using the linear-search-based approach. That is, for a pair of nearby points, denoted by A (for
- which the values of the full sums are already known) and B, the distances from these points to all other points need
- to be calculated. Next, if a distance from some point P to A is smaller than the smoothing kernel radius, and at the
- same time, the P to B distance is larger than the smoothing kernel radius, then the terms concerning point P need to be
- subtracted from the values of sums for A (or added if vice versa is true) to obtain the sums for B.
- For the smoothing to be performed, the only information needed at each point is which previously calculated point is
- used as a reference, and the list of points that need to be added and subtracted.
- Determining the reference points can be performed in a simple manner. First, randomly select the initial point from
- the list of all points this point does not have a reference point since it is the first one. Secondly, select its nearest
- neighbor as the second point and set the first point as its reference. Third, from the list of all remaining unassigned

293 294

298

points, identify the nearest neighbor of the second point and use it as a third point. Fourth, from the list of all the points that have already been assigned (in this case, these are only the first and second points), identify the nearest neighbor and use it as a reference for the third point. Then, repeat steps three and four until all the points have been assigned.

Alternatively, the fourth step could be to always use the point assigned in the previous step as a reference. However, this has some downsides. Namely, the procedure is iterative, with each addition and subtraction incurring a small numerical 283 rounding error. In a field consisting of millions of points, the numerical error could potentially accumulate (especially 284 if a large smoothing kernel is used, as in such cases, the data from hundreds of points might need to be subtracted 285 286 or added at each step). By allowing other than the point assigned in the previous step to be used as a reference, the 287 accumulated numerical error is significantly reduced. There is also a second benefit, namely that the nearest-neighbor search can identify the reference point that is closer and thereby has better overlap of the smoothing kernel than the 288 289 previously assigned point.

Fig. 4a shows the number of iterative steps needed to reach a certain point in the IFS model grid (which consists of about 6.5 million points). As can be observed, the median value is about 19 000 steps, meaning that the number of the required steps is in the tens of thousands, not millions, and thus the numerical error remains limited.

Multiple factors can affect the size of the numerical error. For example, the total number of grid points in the field, the size of the smoothing radius, and which point is selected as the initial starting point. The numerical error will also depend on the nature of the field that is smoothed, for example, whether the original field is less or more variable (like precipitation, which can have large areas with zero values as well as many smaller regions with very large gradients and values). At the same time, even though the error size depends on many factors, the size of the numerical error in a particular setup can be determined relatively easily by comparing the smoothed values obtained via the overlap-detection-based approach to the smoothed values obtained via the kd-tree-based approach, which is as accurate as the linear-search approach and has negligible numerical error. Thus, we recommend that the user first check the magnitude of the numerical error for a few representative fields to make sure it is acceptably small so as not to affect the results of the analysis.

For example, Fig.4b shows the analysis of numerical error for the IFS precipitation field shown in Fig. 2. The graph shows the cumulative distribution of the absolute numerical error (the difference between the smoothed values computed via the overlap-detection and kd-tree-based approaches) for eight different sizes of smoothing radii ranging from 10 to  $15\,000$  km. The graph legend also shows the size of the maximal absolute numerical error for a particular smoothing radius. As expected, the error sizes depend on the smoothing radius, but overall the errors tend to be relatively small, typically smaller than  $10^{-4}$  mm/6h, with the maximum error always smaller than 0.01 mm/6h. Note that this is still substantially less than the typical resolution of the raingauge measurements, which tends to be 0.1 mm or more.

Moreover, although we did not use them here, additional mitigation measures could be implemented to reduce the numerical error further. For example, one could require the explicit calculation of the full sums (over all the points inside the smoothing kernel) each time the number of iterative steps increases by a certain threshold (e.g., every 10 000 steps).

Generating the smoothing data that describes the terms that need to be added or subtracted at each point is relatively slow, but luckily, it only needs to be done once for a particular smoothing kernel size, as it can be saved to disk and then simply loaded into memory whenever needed. For example, generating the smoothing data for the IFS grid for 16 different smoothing kernel sizes (*R* ranging from 10 km to 20 000 km) took about 23 hours when utilizing ten threads.

The smoothing data can take up a lot of space, especially for large smoothing kernels. For example, the data for smoothing a field defined on the IFS grid takes up about 1.2 GB at R=100 km, 12 GB at R=1000 km, and 70 GB at R=10000 km (Fig. 5). At R>10000 km, when the kernel becomes larger than half the Earth's surface area, the amount of data starts to decrease as the length of the border of the smoothing kernel becomes smaller with increasing R due to the spherical geometry of the global domain.

Since smoothing a field using an overlap-detection-based approach requires a simple loop that goes through all the 323 smoothing data while adding or subtracting the appropriate terms, the time complexity of the smoothing calculation 324 is proportional to the size of the smoothing data. Using a single thread, smoothing a field defined on the IFS grid 325 takes about 0.3 s at R = 100 km, 3 s at R = 1000 km, and 45 s at R = 10000 km (Fig. 5). The 15-fold increase in 326 calculation time when R increases from 1000 to 10000 km is larger than one would expect, especially as the increase 327 328 in the size of the smoothing data is only 6-fold. The larger-than-expected increase in computation time is likely related 329 to performance degradation linked to large blocks of memory, which need to be reserved for the smoothing data in case of large smoothing kernels. Likely, the data is split over many RAM modules, which can, in turn, slow down the speed 330 331 of the CPU accessing the data.

Figure 4: (a) A histogram, showing the number of iterative steps needed to reach a specific point for the IFS model grid when using the overlap-detection-based approach. The grid consists of about 6.5 million points. (b) The analysis of numerical error for the overlap-detection-based approach in the case of IFS precipitation field shown in Fig.2. The graph shows the cumulative distribution of the absolute numerical error (the difference between the smoothed values computed via the overlap-detection and kd-tree-based approaches) for eight different sizes of smoothing radii ranging from  $10 \text{ to } 15\,000 \text{ km}$ . The values in the parentheses in the legend show the size of the maximal absolute numerical error, expressed in mm/6h, for a particular smoothing radius.

Figure 5: Size of the smoothing data (green) and computation time (blue) for the smoothing of a field defined on the IFS grid with respect to smoothing kernel radius R using the overlap-detection-based approach. The computation time reflects the time needed on a computer with an AMD Ryzen Threadripper PRO 5975WX processor when utilizing a single thread.

Nevertheless, for efficient calculation, it is best to keep the smoothing data in the memory, where it can be accessed quickly instead of reading it from the disk every time. Thus, it makes sense to load the data from the disk into the memory as part of preprocessing and then use it to smooth multiple fields in a row. The large size of the smoothing data presents a potential problem as it requires the computer to have a large memory, at least in the case of large smoothing kernels that cover a substantial portion of the Earth's surface.

The smoothing calculation can also be parallelized using a shared-memory setup. Namely, the calculation of the sums of the terms that need to be added or subtracted at each point, which represents the computationally most demanding part of the calculation, can be precalculated independently for each point and can thus be calculated in a parallel manner. For example, by using ten threads instead of one, the computation time for R=10000 km reduced from 3 to 0.6 s, while for  $R=10\,000$  km, it reduced from 45 to 12 s. Although the decrease is not tenfold as one would hope (most likely due to the same memory access speed limitations mentioned earlier), the decrease is nevertheless substantial.

# 5 Limited-area domains and missing data

While the focus of this research was the development of methodologies for smoothing of global fields, the approaches presented here can also be used to smooth fields defined on limited-area domains.

Some efficient methods for smoothing fields defined on limited-area domains already exist. For example, the already mentioned summed-fields and Fast-Fourier-Transform-convolution-based approaches (see the Introduction section for

Figure 6: Smoothing in a limited-area domain centered over Europe. The precipitation data is taken from the IFS forecast shown in Fig. 2. (a) the original non-smoothed precipitation field, (b) the smoothed field using a 200 km smoothing kernel radius with the size of the kernel shown with a green circle in the top left corner, (c) the smoothed field using a 200 km kernel radius in the presence of a missing data region in the middle of the domain (indicated in gray).

details). However, the use of these two approaches is limited to regular grids, which are assumed to be defined in a rectangularly shaped domain on a plane, and they also assume equal area sizes for the grid points.

The approaches presented here do not have these limitations and can thus be used with irregular grids defined in non-rectangularly shaped domains, while the smoothed values also reflect the potential differences in area size of different grid points. Moreover, the approaches also correctly handle the spherical curvature of the planet's surface, which can be important in the case of very large domains and for ensuring the consistent size and shape of the smoothing kernel everywhere in the domain.

The smoothing calculation for a limited-area domain is done the same way as for the global domain and is again based on Eq. 1. As before, all that is needed is a list of grid points with values, corresponding latitude and longitude coordinates, and the associated area size data. In the case of a limited-area domain, the points will come only from a specific geographic sub-region, as opposed to the whole Earth, like in the case of a global domain. Any of the two approaches, the k-d-tree-based and the overlap-detection-based, can be used to calculate the smoothed values.

Fig. 6 shows an example of smoothing in a limited-area domain defined over Europe that encompasses the region 20W-40E, 30N-70N. The precipitation data is taken from the IFS forecast shown in Fig. 2, but with points outside the domain removed. Out of 6 599 680 points of the full octahedral reduced Gaussian grid used by the IFS, only 217 421 points inside the domain were selected and used to calculate the smoothed values. Figs. 6a,b show the original and smoothed precipitation using a 200 km smoothing radius.

One noticeable feature is that the values near the domain borders do not decrease towards zero, which would happen, for example, if the smoothing method assumed the values outside the domain were zero. Moreover, although in terms of the latitude/longitude grid, the domain might be considered rectangular, it is not actually rectangular if the spherical shape of the Earth is taken into account, as the northern domain border is much shorter than the southern one.

The methodology presented here can also appropriately handle missing data (i.e., points for which the value is not defined). Frequently, a smoothing method must make some kind of assumption regarding missing data values. For example, the aforementioned summed-fields and Fast-Fourier-Transform-convolution-based approaches must assume some values (e.g., zero is frequently used for precipitation) for the missing data for the calculation of the smoothed values to be successfully performed. This is problematic since it can artificially increase or decrease the values of points close to the regions with missing data (depending on which value is assumed for the missing data points).

With the methodology presented here, the missing data points can be handled appropriately by excluding them from the two sums in Eq. 1. In practice, the same result can be achieved most easily by temporarily setting the area size of these points to zero before proceeding with calculating the smoothed values (this will result in the missing data points not having any influence on the smoothed values).

- Fig. 6c shows an example of smoothing in the presence of missing data, where a region in the center of the domain
- (0E-20E, 40N-50N) was assigned a missing data flag. It can be observed that the values near the missing data region do
- not decrease towards zero, which would happen if the smoothing method assumed the missing data had a zero value.
- In the case of smoothing-based verification, where a pair of fields is compared against each other, and these fields
- contain some missing data points (which are not necessarily at the same locations in both fields), it is best to synchronize
- the missing data (i.e., if a certain point has a missing data flag in one field, then the same point in the other field is 384
- assigned a missing data flag as well) before smoothing is applied to maintain consistency.

### Verification demonstration

- While the main goal of this work was the development of novel smoothing methodologies that are fast enough to be
- used with the output fields of state-of-the-art operational high-resolution global models, we also wanted to include a 388
- limited demonstration of how the developed methodologies could be utilized for smoothing-based global verification.
- Our goal was not to do a proper verification but to showcase how the presented smoothing methodology can be used in
- practice. We chose to focus on the FSS metric since it is one of the most popular spatial verification methods, but the 391
- methodology could easily be used with any other smoothing-based verification metric.
- Thus, we present one example of FSS-based verification of the IFS precipitation forecasts, where the 1-, 3-, 5-, and
- 9-day forecasts of 6-hourly precipitation accumulated between 00-06 UTC on 9 March 2022 are compared against the
- analysis (the precipitation produced in the first 6 hours by the same model initialized on the same day at 00 UTC). We
- note that comparing the forecast against an analysis produced by the same model is problematic in many aspects, but 396
- since our goal was not to do a proper verification but to showcase how the presented smoothing methodology can be
- used in practice, we felt the setup was nevertheless acceptable.
- Fig. 7 showcases the forecast against the analysis fields. As expected, the 1-day forecast (Fig. 7a) exhibits a relatively
- good overlap with the analysis, although some differences can nevertheless be observed, especially in the Tropics. At 400
- longer lead times, the overlap decreases, and the displacements increase as the forecasts become increasingly different
- from the analysis. For example, large displacements are evident in the 9-day forecast (Fig. 7d), especially in the
- mid-latitudes, where large-scale features like cyclones with their fronts and associated precipitation can be substantially
- displaced. 404

386

- To evaluate the forecast performance, we use the original FSS formulation [Roberts and Lean, 2008], which we modify
- to account for different area sizes represented by the grid points

$$FSS = 1 - \frac{\sum a_j (x_j - y_j)^2}{\sum a_j x_j^2 + \sum a_j y_j^2},$$
(3)

where  $x_j$  and  $y_j$  are the values of the smoothed thresholded binary fields at grid point j, and the sums go over all the 407

- points. The  $a_i$  is the representative area size for grid point j. It makes sense that the metric would take into account
- different area sizes since, for example, one would expect a grid point that represents a certain area size to have a smaller
- influence on the metric's value compared to some other grid point that represents twice the area size. If the area size
- is the same for all points (i.e.,  $a_i = a$ ), the FSS expression in Eq. 3 becomes identical to the original formulation in
- Roberts and Lean [2008]. The FSS values can span between 0 and 1, with a larger value indicating a better forecast. 412
- The FSS formulation in Roberts and Lean [2008] nominally uses a square-shaped smoothing kernel/neighborhood but 413
- also mentions the possibility of using other shapes, for example, circular or Gaussian. The use of a square-shaped
- kernel was likely preferable since it is the easiest to calculate in rectangularly-shaped limited-area domains defined on 416
- regular grids in planar geometry. However, using a square-shaped kernel also has a drawback, namely, as the kernel is
- not symmetric, it stretches further in some directions than others (i.e., along the square's diagonal), making the metric 417
- sensitive to the kernel's orientation Skok [2016]. Here we use a sphere-cap-shaped kernel, which is symmetrical and
- correctly takes into account the spherical geometry of the Earth. 419
- Similar to the asymptotic value for the standard FSS (when the neighborhood, aka the smoothing kernel, is large enough
- to cover the whole limited area domain), one can define an asymptotic value for the global domain. As already mentioned
- in Sect. 2, once the smoothing kernel is large enough to cover the whole Earth, the smoothed values will be the same
- everywhere. In this case, the FSS asymptotic value, denoted here as  $FSS_{\rm asy}$ , will be  $1-(x_{\rm asy}-y_{\rm asy})^2/(x_{\rm asy}^2+y_{\rm asy}^2)$ , with  $x_{\rm asy}$  being the asymptotic smoothing values for each field, respectively. Since, in this case, the x and y are 423
- binary fields produced via thresholding, the  $x_{\rm asy}$  and  $y_{\rm asy}$  represent the global frequencies of events in the two fields 425
- (i.e., the portion of the Earth's surface where the original fields have values larger or equal than the threshold). If the 426
- frequency in both fields is the same (i.e.,  $x_{asy} = y_{asy}$ ),  $FSS_{asy}$  will be equal to one and smaller than one otherwise. 427

Figure 7: The visualization of 1-, 3-, 5-, and 9-day IFS model forecasts (red) of 6-hourly precipitation accumulated between 00-06 UTC on 9 March 2022 compared against the analysis (blue).

The FSS value depends on the smoothing kernel size and the threshold. In our case, we used three thresholds roughly

corresponding to low-, medium-, and high-intensity precipitation: 0.1, 1, and 10 mm/6h. We applied the same thresholds

to the whole field. We recognize that using the same threshold for different geographical regions that can have very

diverse climatologies is likely not optimal. Potentially, different climatologically defined thresholds could be used for 432

different regions, but as our main goal was showcasing how to utilize the smoothing methodology, we left this avenue

of exploration for the future.

The results of the FSS-based analysis for the IFS model global precipitation forecasts for the cases shown in Fig. 7 are

visualized in Fig. 8(a-c). The results for the three different thresholds and smoothing radii between 20 and 2000 km are

436 shown.

435

As expected, the FSS values always increase with the smoothing radius. This is expected behavior as using a larger 437

smoothing radius increasingly relaxes the requirement of precipitation events being forecasted at the correct locations. 438

The results are successfully stratified according to the forecast lead time, with the 1-day forecast consistently performing

the best (with the largest FSS value) and the 9-day forecast consistently performing the worst (with the smallest FSS 440

The overall FSS values also decrease with increasing threshold. Namely, at the lowest threshold (0.1 mm/6h, Fig. 8a),

the areas with precipitation tend to be large as they include regions with low-, medium- and high-intensity precipitation.

Since the majority of global precipitation falls in the Intertropical Convergence Zone (ITCZ), the results for the global 444

domain are dominated by the tropics. As the location of the ITCZ changes little on a day-to-day basis, the large regions 445

defined using the lowest threshold exhibit a good overlap, resulting in a relatively high overall FSS value (i.e., mostly 446

larger than 0.7), even at longer lead times. In the mid- and high-latitudes, where large-scale features like cyclones and 447

their fronts can be significantly displaced at longer lead times, the overlap is worse, but since the tropics dominate the

results, the global FSS value is nevertheless high.

On the other hand, regions with more intense precipitation tend to be smaller and exhibit a larger displacement error.

Consequently, their locations in the forecasts overlap less often with their actual locations in the analysis, even in the 451

ICTZ, and the resulting FSS values are lower. For example, the FSS values for the 9-day forecast when using the 10

Figure 8: The FSS- and CSSS-based verification of the IFS model precipitation forecasts shown in Fig. 7. (a-c) The FSS-based verification of global precipitation, (d-f) the CSSS-based verification of global precipitation, and (g-i) the FSS-based verification of the precipitation over the Maritime Continent. The FSS- and CSSS-based verification is done based on Eqs. 3-4.

- 453 mm/6h threshold can be as low as 0.1 (Fig. 8c). Also, the differences in the FSS values between the 1-day and 9-day 454 forecasts are largest at the highest threshold.
- Besides the FSS-based approach described above, we also want a similarly defined smoothing-based metric where 455
- the original non-thresholded fields could be used to calculate the metric's value without thresholding. Namely, using
- the thresholding has some benefits as well as downsides. For example, one benefit of thresholding is that, when 457
- an appropriately high value for the threshold is used, the metric can focus on analysing only the heavy-intensity 458
- precipitation while disregarding the light-intensity precipitation. Another benefit is that the resulting fractions can be
- interpreted in terms of probability (of exceeding the threshold).
- On the other hand, thresholding always removes some information from the field, which can make interpreting the 461
- results more challenging [Skok, 2023]. For example, it does not matter by how much a certain value exceeds the
- threshold the value might exceed the threshold value by just a small amount or a hundredfold the effect on the score's 463
- value will be the same. This issue can be somewhat alleviated by performing the analysis using multiple thresholds, but 464
- this can also make it harder to interpret the results. For example, it can be challenging to determine a general estimate
- of forecast quality for the field as a whole, reflecting precipitation of all intensities. The use of thresholding also makes
- the results sensitive to the selection of the values used for the thresholds, which means a sensitivity analysis needs to be
- performed to determine whether a small change in the thresholds will result in a substantial change in the metric's value.
- Using a metric that does not rely on thresholding avoids some of these issues. 469
- We denote the new metric as Continuous Smoothing Skill Score (CSSS) and define it as: 470

$$CSSS_{p} = 1 - \frac{\sum a_{j} |x_{j} - y_{j}|^{p}}{\sum a_{j} |x_{j}|^{p} + \sum a_{j} |y_{j}|^{p}},$$
(4)

- where  $x_j$  and  $y_j$  are the smoothed values obtained from the original continuous (non-thresholded) fields. Similar to 471
- the FSS, the CSSS values can span between 0 and 1, with a larger value indicating a better forecast. Its asymptotic 472
- value can be expressed as  $CSSS_{p,asy} = 1 - |x_{asy} - y_{asy}|^p/(|x_{asy}|^p + |y_{asy}|^p)$ , with  $x_{asy}$  and  $y_{asy}$  being the asymptotic
- smoothing values for the original non-thresholded fields. If the asymptotic smoothing values of both fields are the same 474
- (i.e., $x_{asy} = y_{asy}$ ),  $CSSS_{p,asy}$  will be equal to one and smaller than one otherwise.
- The p is a user-chosen parameter that influences the score's behavior; more specifically, it defines how much influence
- over the score's value is exhibited by different magnitudes of precipitation intensity. Namely, in the case of p=2 (when
- the CSSS expression is analog to the FSS expression in Eq. 3, but instead of binary fields obtained via thresholding,
- the original non-thresholded fields are used as input), due to the second power, areas with more intense precipitation
- will tend to exhibit a disproportionally large influence on the score's value compared to the areas with less intense
- precipitation. In the case of p=1, this influence will be more proportional, while in the case of p=0.5, the influence
- will again be disproportional, with lower-intensity precipitation exhibiting a comparatively larger influence.
- Fig. 8(d-f) shows an example of CSSS-based verification using p = 0.5, 1, and 2. Same as with FSS, the CSSS 483
- values always increase with the smoothing radius, regardless of the p value. The CSSS results are also successfully
- stratified according to the forecast lead time, with the 1- and 9-day forecasts consistently performing the best or worst,
- respectively. The results for p=2 (Fig. 8d, which is most similar to FSS in terms of how it is defined since an operator
- using the second power is used for both) are most similar to the FSS results for threshold 1 mm/6h. In this case, there is
- a significant difference in the CSSS values for different lead times at a smaller smoothing radius, but this difference
- vanishes at a larger radius when the 1- and 9-day have very similar CSSS values.
- Interestingly, for p = 1 and 0.5 (Figs. 8(e-f)), the difference between the results for lead times tends to be somewhat 490
- smaller (compared to p = 2), but it persists at all smoothing radii, indicating that 1-day forecast outperforms the 9-day
- forecasts even if a very large smoothing radius is used. 492
- To further examine the effect of the p parameter on the results, we calculated some relevant quantities using the analysis
- field, shown in Table 1.
- The table shows the portions of the surface area, precipitation volume, and the  $\sum a_i |x_i|^p$  sums from the denominator
- of Eq.4 for different latitudinally-defined regions, namely the Tropics (30S-30N), the Midlatitudes (30S-60S and 496
- 30N-60N), and the Polar regions (60S-90S and 60N-90N). The  $\sum a_j |x_j|^p$  for a particular region can be used as a kind 497
- of rough indicator of how much influence on the CSSS values is exhibited by the region. It depends on the area size of 498
- the region and the amount and intensity of precipitation found in it, as well as on the value of the p parameter.
- For example, the Tropics cover about 50% of the Earth's surface and contain about 56% of the total precipitation volume
- in the analysis field, but nevertheless contribute about 62% to the sum for p=2. On the other hand, the Polar regions 501
- cover about 13% of the Earth's surface and contain about 6% of the total precipitation volume but contribute less than
- 2% to the sum for p=2. This highlights how the Tropics have a disproportionately large influence on the resulting 503
- values at the expense of other regions, especially the Polar regions, which exhibit almost no influence on the result. 504

Table 1: The portions of the Earth's surface area, total precipitation volume, and the  $\sum a_j |x_j|^p$  sums from the denominator of Eq.4 for different latitudinally-defined regions: the Tropics (30S-30N), the Midlatitues (30S-60S and 30N-60N), and the Polar regions (60S-90S and 60N-90N). The values were calculated for the original (non-smoothed) analysis field of the IFS forecast shown in Fig. 7.

|   | Region        | Surface | Precipitation | $\sum a_j  x_j ^p$ |       |         |
|---|---------------|---------|---------------|--------------------|-------|---------|
|   |               | area    | volume        | p=2                | p=1   | p = 0.5 |
| - | Tropics       | 50.0%   | 56.4%         | 62.3%              | 56.4% | 51.8%   |
|   | Midlatitudes  | 36.6%   | 37.4%         | 35.9%              | 37.4% | 37.7%   |
|   | Polar regions | 13.4%   | 6.2%          | 1.8%               | 6.2%  | 10.5%   |

- The situation is different for p = 0.5, where the Tropics contribute about 50% to the sum, with the Polar regions contributing about 10%, meaning that they can have a noticeable influence on the result. This highlights how the p parameter can be used to adjust the comparative influence of drier and wetter regions.
- Finally, although global forecast quality information can be useful, the information for specific geographic sub-regions (such as a continent, a country, or a latitude belt) or types of surfaces (like land or sea) is often of greater interest, even in cases when a global model is used. In this case, smoothing can be performed for the whole global field, as before, but only a subset of smoothed values can then be used to determine the forecast quality for a specific sub-region. Smoothing
- the whole global field first avoids the so-called border-effect issues that can arise while using a smoothing-based metric
- in a limited area domain. For example, in some cases, the FSS value can markedly decrease or increase in a limited area
- domain depending on how the domain border is handled [e.g., Skok and Roberts, 2016]
- Fig. 8(g-i) shows FSS computed over the Maritime Continent. Smoothing is still performed globally, but the FSS score
- is computed via Eq. 3 by using only the grid points that fall between latitudes 15°S and 15°N and longitudes 90 and
- 150°E.

526

- The FSS score is quite high for the low and medium thresholds (Figs. 8(g-h)). This happens since the precipitation over
- the Maritime Continent mostly occurs in the ITCZ, which does not move much on a daily basis. Thus, the location
- of the medium-, and especially low-intensity precipitation envelopes that surround the convective cores containing
- high-intensity precipitation and cover a large area do not move much. This makes the forecast quality of lower-intensity
- precipitation almost independent of forecast lead time (e.g., the 1-day forecast is almost as good as the 9-day forecast).
- The situation is different for high-intensity precipitation, as the model can frequently struggle to correctly forecast the
- positions and intensity of precipitation in the convective cores, especially at longer lead times. Consequently, the FSS
- score is lower and more variable in this case (Fig. 8i).

# 7 Discussion and Conclusions

- We present two new methodologies for smoothing fields on a sphere that can be used for smoothing-based verification
- in a global domain. One is based on k-d trees and one on overlap detection. The k-d-tree-based approach requires less
- memory, has negligible numerical error, and can be done in a single step without any additional preprocessing, but it is
- slower, especially for large smoothing kernels.
- The overlap-identification-based approach requires a preprocessing step that generates the smoothing data, which must
- be calculated only once for a specific smoothing kernel size. Once available, this data can be used to calculate the
- smoothed values much faster than the k-d-tree-based approach. The large size of the smoothing data presents a potential
- problem as it requires the computer to have a large memory (this is only problematic if a very large smoothing kernel is
- used). Since the procedure is iterative, the approach can also incur a degree of numerical error, but luckily, the size of
- the numerical error in a particular setup can be determined relatively easily by comparing the smoothed values obtained
- via the overlap-detection-based approach to those obtained via the kd-tree-based approach. Moreover, simple mitigation
- strategies exist that can be implemented to reduce the error size further.
- Alternatively, similarly to how it is done for the overlap-identification-based approach, the smoothing data for the
- k-d-tree-based approach, which would list all the nodes that need to be summed to get the smoothed value at a specific
- location for a particular size of the smoothing kernel, could be precalculated and saved to a disk. This data could then
- be simply loaded into memory when needed and used to quickly calculate the smoothed values, similarly to how it is
- done for the overlap-identification-based approach. However, testing showed that the size of this data is a few times
- larger than for the overlap-identification-based approach, meaning that the calculation of the smoothed values would be
- correspondingly slower.

- Both methodologies can be used when the grid is not regular, thus avoiding the need for prior interpolation into a
- regular grid, which can introduce additional smoothing [Konca-Kedzierska et al., 2023]. They also take into account
- the spherical geometry of Earth, which is important to ensure a consistent size and shape of the smoothing kernel
- everywhere on the planet.
- The methodologies are also area-size-informed, meaning that they take into account the potentially different area sizes
- of the grid points. This is important since in some grids (e.g., a regular latitude/longitude grid), the difference between
- the area sizes of points at different locations on the planet can be very large. Not accounting for this could result in
- negative effects, for example, the spatial integral of the field could change considerably due to the smoothing.
- While the focus was on the development of methodologies for smoothing of global fields, both approaches can also be
- used in limited-area domains. Moreover, they are both able to deal with missing data appropriately. This is important
- since dealing with missing values can be problematic for some smoothing methods, as they are often forced to make
- some kind of assumptions regarding the value of missing data, which can cause the values near the missing data region
- to increase or decrease artificially.
- Overall, while each approach has its strengths and weaknesses, both are potentially fast enough to make the smoothing
- of high-resolution global fields feasible, which was the primary goal set at the beginning. The time complexity of both
- approaches can be approximated by  $O(n\sqrt{k})$  with k being the typical number of points in the smoothing kernel, which
- is limited by n in the worst case.
- Based on the methodologies presented here, we prepared and published an easy-to-use Python software package for
- efficient calculation of the smoothing (please refer to the Code and data availability statement for details on how to
- obtain the package).
- In addition to the novel smoothing methodologies, we also included a verification demonstration where we presented an
- area-size-aware variant of the FSS, which takes into account the varying area sizes that are representative of different
- grid points. For example, one would expect a grid point with a larger area size to exhibit a larger influence on the
- metric's value compared to one with a smaller area size. We also defined a smoothing-based metric, the CSSS, where
- the original non-thresholded fields can be used to calculate the metric's value without thresholding. The CSSS has a
- user-selectable exponential parameter that affects how the precipitation magnitude influences the value of the metric,
- which can be used to adjust the comparative influence of drier and wetter regions. We also demonstrate how the
- smoothing-based scores can be used to provide localized forecast quality information for a global forecast by first
- smoothing the fields globally, thereby avoiding the border-effect issues that can arise for limited area domains, and
- then using a regionally-defined subset of points to calculate the metric's values representative of a specific sub-region.
- Alternatively, one could also obtain even more localized information of forecast quality, for example, by calculating the
- Localized Fraction Skill Score [LFSS, Woodhams et al., 2018]. In this case, the fraction values in the global domain
- can be calculated efficiently using the new smoothing methodology, in the same way as before, but then, the fraction
- values can be used to calculate the LFSS instead of the "regular" FSS.

#### 580 Funding

- Slovenian Research And Innovation Agency (Javna agencija za znanstvenoraziskovalno in inovacijsko dejavnost RS)
- research core funding No. P1-0188.

#### 583 Author Contributions

- **GS**: conceptualization, data curation, formal analysis, funding acquisition, investigation, methodology, resources,
- software, validation, visualization, writing original draft, writing review & editing. **KK**: investigation, validation,
- writing original draft, writing review & editing.

# 587 Acknowledgements

- The authors are grateful to Llorenç Lledó and Willem Deconinck from ECMWF for kindly providing the sample
- fields of IFS model precipitation forecasts, along with the grid point area size data, that were used to demonstrate the
- smoothing methodology.

## 591 Code and data availability

- We prepared and published an easy-to-use Python software package for efficient calculation of the smoothing on the
- sphere. The underlying code is written in C++, and a Python ctypes-based wrapper is provided for easy use within
- the Python environment. The package contains all the source code as well as some examples and sample fields that
- are used to demonstrate its usage. The current version of the package is available at https://github.com/skokg/
- Smoothing\_on\_Sphere under the MIT license. A snapshot of the version of the package that was used to calculate
- the results presented in this paper, which additionally includes all the sample fields that were used as input data, is
- archived on Zenodo repository under DOI 10.5281/zenodo.15087716 [Skok, 2025]. The C++ code uses float64 with the
- exception of the code for k-d tree construction and nearest neighbor search, which uses float32 to increase computational
- speed.

#### 601 Conflict of Interest Statement

The authors declare no conflicts of interest.

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
