# Peer review of "SMOOTHING AND SPATIAL VERIFICATION OF GLOBAL FIELDS"

_EGUsphere, 2025_

## Author Comment (AC1)

We want to thank both reviewers for a thorough review and the valuable comments and suggestions on improving the manuscript's quality. As a result, notable improvements were made to the manuscript. We also made a number of smaller changes to the text to try to improve the language and clarity.

Below, we provide replies (blue colored text) to the specific points raised by the reviewers.

**Reviewer 1**

The manuscript presents two strategies for neighborhood-based smoothing. A kd tree range-search and an "advanced-front" style overlap-detection with reused intermediate steps on high-resolution grids on the spherical surface. While the topic is important and the ideas are promising, several gaps need to be closed before publication.

1. Clarify grid and geometry assumptions: When discussing the grid used in the model, please include detailed definitions, such as the grid data format used in the paper. Does it require connectivity or not? An example way to say this is: We only use a grid point for each face, without connectivity information. For each grid face, we just assume the face area is known. And when talking about a spherical surface, indicate if the sphere is a unit sphere or the sphere with the Earth's radius. And also, how did you handle the spherical geometry problem while you already assumed the grid area is provided?

Thank you for this suggestion. We have added additional information regarding the grid and geometry assumptions, as well as about the IFS model and its grid. The smoothing methodology does not require the connectivity information. We assumed the sphere radius was equal to the Earth's radius. The grid area data for the IFS model grid was provided by the ECMWF. We are not sure how they calculated it, but it is the same data they use for their analysis. If the area size data were unavailable, it could be calculated by performing the Voronoi tessellation on the sphere, for example.

Consequently, the new version of the relevant text in the Area-size-informed smoothing section had changed to:

*"Fig. 2 shows some examples of smoothed fields of forecasted 6-hourly accumulations of precipitation produced by the high-resolution deterministic Integrated Forecasting System [IFS, ECMWF, 2023a,b] of the European Centre for Medium-Range Weather Forecasts (ECMWF). The IFS is considered one of the best-performing operational medium-range global deterministic models and is frequently used as a benchmark to which other models are compared against [e.g., Bi et al., 2023, Lam et al., 2023, Lang et al., 2024], which makes it especially suitable to be used as an example.*

*The IFS uses an octahedral reduced Gaussian grid O1280 [Malardel et al., 2016], which consists of around 6.5 million grid points. The points are arranged in fixed-latitude circular bands, with the band closest to the equator consisting of 5136 equidistant points spread around the Earth. In the poleward direction, each next band has four points fewer than the previous one, with the last band, located close to the poles, consisting of only 20 points. This setup makes the grid irregular, with area size of the points also varying substantially with latitude, from 61~km^2 at the equator, to 93~km^2 at 75º, where it is the largest, to 18 km^2 close to the poles, where it is*

*the smallest t [Skok and Lledó, 2025. The IFS precipitation data was provided to us by the ECMWF in the form of netCDF files that contained the lat-lon locations of the points, the precipitation accumulation values, and the area size data of all the points. All the numeric data was provided in float32 numeric format.*

*The smoothing methodology represented by Eq.1 does not have any limitations or requirements about the grid being regular - the only assumption is that the points are located on a sphere (in our case, we also assumed that the sphere radius was equal to the Earth's radius). It is worth noting that the smoothing methodology does not require the connectivity information. The only data required for calculating the smoothed values are the original field values, the locations of all the points on the sphere, and their area size information. In the case of IFS fields, the area size data was already provided by the ECMWF, but if it was not available, it could be obtained by performing the Voronoi tessellation on the sphere, for example."*

2. Consider re-using the kd-tree during overlap pre-processing:

The manuscript treats the two algorithms as mutually exclusive: k-d tree for "small/medium" radii and overlap-detection for "large" radii. But the overlap method already needs to enumerate every neighbourhood during its one-time table build, and it's in O(n) time complexity. So, intuitively, building the O(nlogn) k-d tree for this overlap step is beneficial, and it can reduce the neighbour search as well. Since it keeps the cap-membership logic identical in both schemes, it can also reuse the previous structure. So, a discussion point for such a hybrid method will be helpful. Even if you decide not to proceed with the hybrid method, some sentences explaining what the limits are will help to comprehensively discuss this problem. Finally, giving some benchmark experiments showing when it's a good time for the k-d tree algorithms and when it's a good time for the overlap detection will be beneficial as well.

Thank you for this suggestion - we hope we understand it correctly. During the preparation of the smoothing data for the overlap-based approach, the k-d tree is already constructed. Constructing the k-d tree is quite fast – it only takes a few seconds for the ~6.5 million points of the IFS grid. The tree is then used to determine the iterative sequence of points, which requires many nearest neighbour searches. We also tried using the k-d tree to identify the overlaps themselves, but it turned out that this does not work so well. Namely,  as opposed to searching only for the closest nearest neighbour (which has O(log(n)) cost, with n being the total number of grid points), identifying all the points inside a specific search radius has O(log(n)*k) cost, where k is the typical number of points inside the radius. Since n such searches need to be made, the total cost is O(n*log(k)*k). For small smoothing kernels (when k << n), this will be faster than the linear-search-based approach, but for bigger kernels, as k becomes larger, the benefit vanishes. Moreover, once the lists of all the points inside a search radius are identified for both the current and the reference point, the difference between the two lists needs to be determined to identify which points are in one list but not the other (and vice versa). It turns out both lists need to be sorted for the differences to be identified efficiently, with the sorting nominally being an O(k*log(k)) operation. Since this needs to be performed for every point, the total cost of this step is O(n*log(k)*k), which again becomes expensive for larger smoothing kernels. This is why we decided to keep using the linear search for this step. We do not consider this a big issue, since this step needs to be performed only once for a particular search radius. Moreover, our code supports the generation of smoothing data for multiple radii in a single calculation – the approach reuses some results of the interim calculations for different radii,

thereby reducing the overall computational cost (compared to the data being generated for each radius separately).

3. More general numerical-error analysis is required for a broader audience. The statement that overlaps deviates by "< 0.01 mm / 6 h" lacks context. Some readers do not work with precipitation. Add a paragraph that (i) defines the brute-force/k-d result as the reference and (ii) explains why that is the correct ground truth for round-off studies. Provide information on absolute error/relative error/ maximum error across representative radii instead of using the term "< 0.01 mm / 6 h". An error-growth plot versus traversal depth (0,2000,400,..., 20000 hops) will also be helpful.

We added more information and discussion regarding the numerical error. We also prepared a new figure (Figure 4b in the latest version of the manuscript) showing a more detailed analysis of the numerical errors for the precipitation field in the example, by plotting cumulative distributions of the absolute error as well as showing the maximum error at multiple smoothing radii. We did not analyse relative error since it is problematic in the case of precipitation as the precipitation value can frequently be equal to zero, and even a very small error in absolute terms would result in an infinitely large relative error due to division by zero.

We show the new figure and the associated text below:

[Figure]

*"Figure 4: (a) A histogram, showing the number of iterative steps needed to reach a specific point for the IFS model grid when using the overlap-detection-based approach. The grid consists of about 6.5 million points. (b) The analysis of numerical error for the overlap-detection-based approach in the case of IFS precipitation field shown in Fig.\ref{fig:IFS_smoothing_examples}. The graph shows the cumulative distribution of the absolute numerical error (the difference between the smoothed values computed via the overlap-detection and kd-tree-based approaches) for eight different sizes of smoothing radii ranging from 10 to $15\,000$~km. The values in the parentheses in the legend show the size of the maximal absolute numerical error, expressed in mm/6h, for a particular smoothing radius."*

The new text:

*"Multiple factors can affect the size of the numerical error. For example, the total number of grid points in the field, the size of the smoothing radius, and which point is selected as the initial starting point. The numerical error will also depend on the nature of the field that is smoothed,*

*for example, whether the original field is less or more variable (like precipitation, which can have large areas with zero values as well as many smaller regions with very large gradients and values). At the same time, even though the error size depends on many factors, the size of the numerical error in a particular setup can be determined relatively easily by comparing the smoothed values obtained via the overlap-detection-based approach to the smoothed values obtained via the kd-tree-based approach, which is as accurate as the linear-search approach and has negligible numerical error. Thus, we recommend that the user first check the magnitude of the numerical error for a few representative fields to make sure it is acceptably small so as not to affect the results of the analysis.*

*For example, Fig.4b shows the analysis of numerical error for the IFS precipitation field shown in Fig.2. The graph shows the cumulative distribution of the absolute numerical error (the difference between the smoothed values computed via the overlap-detection and kd-tree-based approaches) for eight different sizes of smoothing radii ranging from 10 to 15000 km. The graph legend also shows the size of the maximal absolute numerical error for a particular smoothing radius. As expected, the error sizes depend on the smoothing radius, but overall the errors tend to be relatively small, typically smaller than $10^{-4}$ mm/6h, with the maximum error always smaller than 0.01 mm/6h. Note that this is still substantially less than the typical resolution of the raingauge measurements, which tends to be 0.1 mm or more.*

*Moreover, although we did not use them here, additional mitigation measures could be implemented to reduce the numerical error further. For example, one could require the explicit calculation of the full sums (over all the points inside the smoothing kernel) each time the number of iterative steps increases by a certain threshold (e.g., every 10000 steps)."*

We also expanded the text a bit in the Discussion and Conclusions sections, where we discuss the numerical error, to make it more clear how the size of the numerical error can be determined by comparing the smoothed values obtained via the overlap-detection-based approach to those obtained via the kd-tree-based approach. The last two sentences of the relevant paragraph have thus been changed to:

*"Since the procedure is iterative, the approach can also incur a degree of numerical error, but luckily, the size of the numerical error in a particular setup can be determined relatively easily by comparing the smoothed values obtained via the overlap-detection-based approach to those obtained via the kd-tree-based approach. Moreover, simple mitigation strategies exist that can be implemented to reduce the error size further."*

4. Providing fuller details for the experiment setup will be helpful: What kind of grids of the experiments, and why can they represent the common grid types? How do these grids reveal the nature of the spherical surface? And when carrying out experiments, are you consistently using float64? A clear definition of the experiment setup will greatly help.

We added more information regarding the setup we used in the study. For example, a more extensive description of the IFS model grid was added (please refer to our response to point 1). The IFS precipitation data was provided to us in float32 format (we added this information to the text). The C++ code of the smoothing library we prepared consistently uses float64 except for k-d tree construction and nearest neighbour search, which use float32 to increase computational speed - we added this information to the Code and data availability section, where we added a sentence: "*The C++ code uses float64 with the exception of the code for k-d tree construction and nearest neighbor search, which uses float32 to increase computational speed*".

We also added additional information on our computational setup (into the part of the text where we describe IFS model and the grid in section "Area-size-informed smoothing"). The text we added is:

*"Our computational setup consisted of a computer with an AMD Ryzen Threadripper PRO 5975WX processor with 32 physical cores. The Debian 12 Linux operating system was installed on the computer. The code was written in C++, and the gcc compiler version 12.2 was used to compile the code with the OpenMP programming interface used for shared-memory multi-thread computing. Hyper-threading was enabled. Even though the IFS data was provided in float32 format, we consistently used double (float64) precision in the C++ code, except in one special case (for more information, please refer to the "Code and data availability" section)"*

Technical corrections:

p. 8, Fig. 5 caption: add CPU model and numeric precision.

We are not sure what is meant by the CPU model – we already specified the CPU is AMD Ryzen Threadripper PRO 5975WX, and as far as we are aware, there is only one model of this CPU. We added more information regarding the float precision we used in our code into the "Area-size-informed smoothing" section, where we describe our computational setup (please refer to our response to the previous point).

Throughout, when stating runtimes ("single core", "ten cores"), clarify whether these refer to physical cores or hardware threads, and whether hyper-threading was enabled. This will help readers reproduce the scaling results.

Thank you for pointing this out. Upon inspection, we realized that we specify the number of threads, not cores, in the OpenMP programming interface. Hyper-threading was enabled on the system. Since the maximum number of threads we used was 10, while the processor has 32 physical cores, each thread should, in principle, be assigned to its own physical core. Still, as this is not guaranteed, we changed "core" -> "thread" throughout the manuscript to be more accurate. When describing our computational setup, we also clarified that hyper-threading was enabled.

**Reviewer 2**

Speeding up computationally expensive algorithms is a highly desirable thing, and the analysis on the algorithm options and their computational costs is very interesting and useful to know to improve. The "need for speed" when crunching through 5 km global grids is undeniable. However, I did not feel the paper had the right title. The paper tries to argue for a lot of complexity. Unfortunately, I was not convinced of the necessity. This is perhaps because of the focus on precipitation, which at a global scale, is a very tricky prospect.

From a scientific/philosophical perspective I have a number of comments to make.

1. My philosophical point is the whole reference to smoothing. You start by showing the smoothing of precipitation fields, but for the FSS the precipitation fields are NOT smoothed. It is the process of thresholding and computing fractions over increasingly large neighbourhoods that provides the smoothing. I would not use these methods to coarse grain a raw field. I would probably regrid using a conserving regridder.

We acknowledge that in the case of the FSS, the fields are first thresholded, and then the resulting binary fields are used to calculate the fractions inside a neighbourhood of prescribed size, which are then finally used to calculate the score's value. An important insight here is that the calculation of the fraction values from the binary fields is mathematically equivalent to the smoothing of the binary field. At the same time, this is the computationally most demanding step in the calculation of the FSS value, which can be especially difficult/costly to perform in spherical geometry where the grid will be inherently non-equidistant and/or irregular. We also note that the calculation of fractions/smoothing is not the same as coarse-graining/regridding of a field. In case of calculation of fractions/smoothing the resolution and grid of the resulting field must be identical to the original field, which is not the case in case of coarse-graining/regridding. In the latest version of the manuscript, we tried to present more clearly the link between fraction calculation and smoothing, as it was perhaps not highlighted clearly enough in the previous version. The new version of the relevant text in the Introduction was changed to:

*"The Fraction Skill Score [FSS, Roberts and Lean, 2008, Roberts, 2008] is a widely used neighborhood-based verification metric. It works by first applying a threshold, thereby converting the original fields to binary fields, and then calculating the fractions that represent the ratio between the number of non-zero and all points located inside a neighborhood of prescribed shape and size, which are then used to calculate the score's value. We note that calculating the fraction values from a binary field is mathematically equivalent to smoothing the binary field using a constant value smoothing kernel of the same shape and size as the neighborhood."*

Regarding the use of non-thresholded fields for verification, while the FSS is an example of a spatial verification metric that does use thresholding, many other metrics work with the original (non-thresholded) fields. Some examples are, the smooting-based PSD metric (https://doi.org/10.3390/app12084048), the wavelet-based SAD (https://doi.org/10.1002/qj.2881) and optical-flow-based DAS (https://doi.org/10.1175/2009WAF2222247.1).

In relation to focus on smoothing, the main goal of this work was to develop computationally efficient methodologies for smoothing on the sphere, which could be used with fields defined on any kind of grid, would be area-size aware, and would be able to handle the spherical geometry correctly as well as any missing data. As far as we know, no such methodologies have yet been developed, and we are the first. And, while the motivation for the work does originate from the verification, the developed methodology could, in the future, perhaps also be useful for researchers who might need to use smoothing on the sphere for their work on other topics, whatever they might be.

2. In lat-lon space these grids are regular, and new cube-sphere global grids are also regular, in physical spacing too. There is nothing wrong with performing global verification on a regular lat-lon grid. The issue comes with aggregation and interpretation. And here you are mixing apples and oranges when considering the combination of grid points at 45N and 15N. This is where real distances become more problematic. Forget about anything north of 45N or so. We have no reliable gridded rainfall analyses that are any good. Fundamentally I have problems with computing a single score for global precipitation spanning large regions, e.g. NH, TR and SH, but even smaller regions. Europe has an extremely heterogeneous precipitation climate. As papers on this subject, e.g. SEEPS (Rodwell et al, 2010; Haiden et al. 2012, North et al. 2022) demonstrated, the precipitation climatology globally varies so much, even at the same latitude, that some form of local climatology must be used to verify precipitation to not fall foul of false skill (a la Hamill and Juras, 2006).

We agree that there is nothing wrong with performing global verification on a regular lat-lon grid – as long as the spherical geometry of the Earth is properly taken into account (but this can often be difficult). For example, in the case of FSS, we are unaware of any computationally efficient examples of calculating the fractions in high-resolution global fields (while correctly taking into account the spherical geometry) for all but the smallest neighborhoods.

We are not experts on cube-sphere grids, but we have reservations about believing that cube-sphere global grids can be fully regular in physical spacing – for example, a fully regular grid in physical spacing would mean that the area sizes and great-circle distances to nearby points would all be the same for all grid points in the grid. Generally, one cannot assume most of the global models will have regular grids in the future. For example, the currently operational IFS model used in this study uses a octahedral reduced Gaussian grid that has the points arranged in fixed-latitude bands, but in the poleward direction, each next band has four fewer points than the previous one, resulting in the grid being irregular (per suggestion of the other reviewer, we also added a more detailed description of the IFS grid to the manuscript). Moreover, the new global machine-learning-based models developed and used by some operational centers and big data companies frequently also rely on grids that are not regular. For example, the GraphCast model developed by Google is based on a graph neural network that uses icosahedral mesh (https://doi.org/10.1126/science.adi2336), while the ECMWF's AIFS uses an octahedral reduced Gaussian grid (https://doi.org/10.48550/arXiv.2406.01465). This is why we focused on developing a general smoothing methodology that can work with any kind of grid.

We agree that no accurate global precipitation datasets exist, with the errors of the available datasets being especially large at high latitudes. Nevertheless, a need for such datasets still exists, even though they might not be very accurate. One example is the new global machine-learning-based models. To be able to forecast precipitation in a global domain, such models

need to be trained on the global precipitation data – there does not seem to be any other viable alternative.

We acknowledge the potential issues when calculating a single global score value for a climatically diverse variable such as precipitation (for example, in the manuscript, we mention the issue of using the same thresholds for all regions when doing the FSS-based analysis). Although there are some cases when having a single global value is useful or necessary (e.g., if one would like to compare the general performance of two global models for the whole global domain or if the verification metric would be used as a loss function during training a global machine-learning-based model), in most cases, focusing on local estimates of forecast quality would make more sense. Luckily, the smoothing methodology was designed to be general enough and can thus be easily utilized for this purpose – for example, to calculate the FSS or CSSS for a smaller region (e.g., like the Maritime Continent region shown in the manuscript), or to calculate the localized FSS (LFSS) as defined by Woodhams et al. (https://doi.org/10.1175/MWR-D-17-0396.1). In the case of LFSS, the fraction values in the global domain can first be calculated efficiently using the new smoothing methodology, but then, instead of calculating the "regular" FSS, the fraction values can be used to calculate the LFSS. In the latest version of the manuscript, we added some information about the possibility of using the smoothing methodology to calculate the LFSS. We added the following text into the Discussion and Conclusions section:

"*Alternatively, one could also obtain even more localized information of forecast quality, for example, by calculating the Localized Fraction Skill Score [LFSS, Woodhams et al., 2018]. In this case, the fraction values in the global domain can be calculated efficiently using the new smoothing methodology, in the same way as before, but then, the fraction values can be used to calculate the LFSS instead of the "regular" FSS.*"

3. The authors introduce the CSSS. This feels to me like a bit of an afterthought. If I understand this correctly, this is not relying on neighbourhoods or thresholds but does require the raw field to be smoothed. What would be the benefit of this over just regridding using a conserving regridder? Is it speed?

The comment touches on two different aspects. One is the use of thresholding vs. non-thresholding, and the other is the use of fractions/smoothing vs regridding.

With regard to the use of thresholding – we acknowledge that we did not explain clearly enough the motivation behind the CSSS in the previous version of the manuscript – thank you for pointing this out. The FSS and CSSS are conceptually very similar and defined almost identically in mathematical terms – the main difference is that the FSS requires the input fields to be thresholded while the CSSS does not (a secondary difference is that the CSSS has an additional exponent parameter that is set by the user). Generally, neither of the two approaches (using or not using thresholding) is inherently worse or better than the other, but they are different, and each has some benefits as well as downsides. For example, one benefit of thresholding is that, when an appropriately high value for the threshold is used, it can focus on analysing only the heavy-intensity precipitation while disregarding the light-intensity precipitation. Another benefit is that the resulting fractions can be interpreted in terms of probability (of exceeding the threshold).

On the other hand, thresholding always removes some information from the field, which can make interpreting the results more challenging. For example, it does not matter by how much a

certain value exceeds the threshold – the value might exceed the threshold value by just a small amount or a hundredfold - the effect on the score's value will be the same. This issue can be somewhat alleviated by performing the analysis using multiple thresholds, but using many thresholds can also make it harder to interpret the results. For example, it can be challenging to determine a general estimate of forecast quality for the field as a whole, reflecting precipitation of all intensities. The use of thresholding also makes the results sensitive to the selection of the values used for the thresholds, which means a sensitivity analysis needs to be performed to determine whether a small change in the thresholds will result in a substantial change in the metric's value. By not relying on thresholding, the CSSS avoids some of these issues. For example, it is not sensitive to the selection of the value for thresholding and can produce a general estimate of forecast quality reflecting precipitation of all intensities, with the exponent parameter affecting how much influence will be given to different intensities. This is why we believe it can be useful, but at the same time, we do not suggest the CSSS is inherently better or should completely replace the FSS, as each metric informs on somewhat different aspects of forecast quality.

To make the motivation behind CSSS clearer we added additional explanations to the manuscript. The new version of the relevant text is:

*"Besides the FSS-based approach described above, we also want a similarly defined smoothing-based metric where the original non-thresholded fields could be used to calculate the metric's value without thresholding. Namely, using the thresholding has some benefits as well as downsides. For example, one benefit of thresholding is that, when an appropriately high value for the threshold is used, the metric can focus on analysing only the heavy-intensity precipitation while disregarding the light-intensity precipitation. Another benefit is that the resulting fractions can be interpreted in terms of probability (of exceeding the threshold).*

*On the other hand, thresholding always removes some information from the field, which can make interpreting the results more challenging (Skok, 2023). For example, it does not matter by how much a certain value exceeds the threshold – the value might exceed the threshold value by just a small amount or a hundredfold - the effect on the score's value will be the same. This issue can be somewhat alleviated by performing the analysis using multiple thresholds, but this can also make it harder to interpret the results. For example, it can be challenging to determine a general estimate of forecast quality for the field as a whole, reflecting precipitation of all intensities. The use of thresholding also makes the results sensitive to the selection of the values used for the thresholds, which means a sensitivity analysis needs to be performed to determine whether a small change in the thresholds will result in a substantial change in the metric's value. Using a metric that does not rely on thresholding avoids some of these issues. "*

Concerning the use of regridding instead of using fractions/smoothing. As mentioned, using fractions/smoothing is not mathematically equivalent to regridding. While scores similar to FSS or CSSS could potentially be defined using regridding, they would not be the same scores, and their behaviour would likely be different in ways that are not immediately obvious, and would depend on the exact method used for the regridding.

4. A general sense of dissatisfaction, I am led to believe, in not being able interpret/translate back what the FSS means, in terms of how to improve the model, was a primary motivation behind moving to the localised version of the FSS (Woodhams et al) and whilst the authors Woodhams et al. and Mittermaier may disagree on the size of the neighbourhoods used, the latter also demonstrated that high scores do not necessarily indicate skill because persistence

scores even higher (over the Maritime Continent). That study has flaws, as you point out, but my counterargument would be that if one is not trying to aggregate over large regions (and there are very good reasons why one shouldn't without accounting for climatology), then the issue of what the underlying grid is becomes far less important. Furthermore, the FSS may be popular because it is easy to compute but that does not mean it is that helpful in differentiating skill, as Antonio and Aitchison (2024) also recently demonstrated.

In general, no single verification metric exists that would be perfect and work the best for every purpose and in every situation. Each metric has its strengths and weaknesses, and it typically reflects only some aspects of the forecast performance – this is why it is essential to always use multiple different metrics to evaluate the forecast performance, instead of relying on only one.

The issue of interpretability is not limited to FSS, as it is also true for many other verification metrics. Moreover, even when the results can be interpreted in some comprehensible manner, there is no guarantee that it will be obvious in what way the model should be changed to improve the forecast.

The FSS is indeed one of the most popular metrics, but at the same time, its behaviour and properties have likely also been one of the most analysed and scrutinised by various researchers. The study by Antonio and Aitchison is just one of the latest examples. In their accepted version of the paper, which was published in the Monthly Weather Review just a few months ago (https://doi.org/10.1175/MWR-D-24-0120.1), they focus on trying to provide a new and more robust method to determine forecast skill from the FSS (by using random forecast as a reference), as opposed to simply saying that FSS is not good and discouraging its use. FSS is by no means perfect, no metric is, but it continues to be developed and used, and not necessarily only due to being easy to compute (nowadays, most researchers likely use one of the freely available verification libraries to compute the FSS, but those same libraries make it easy to calculate all kinds of metrics, not just FSS). Even more importantly, FSS spurred the development of many derivative metrics, as different researchers tried to improve or extend its functionality by developing metrics that can deal with ensemble forecasts, analyse timing errors and variables other than precipitation, and provide localized estimates of forecast quality. Most of these metrics are based on the same underlying principles, meaning that the presented smoothing methodology could help with their efficient calculation when the spherical geometry needs to be considered.

Another potential use of spatial verification metrics is that they can be used as a loss function for training the machine-learning-based models. There had not yet been many studies on this topic, but one pioneering study used multiple spatial metrics for this purpose and found that FSS was one of the best-performing metrics in this regard (Lagerquist & Ebert-Uphoff, 2022, https://doi.org/10.1175/AIES-D-22-0021.1).

From a verification "best practice" perspective, I am reluctant to see studies on global precipitation forecast verification published which are not accounting for the peculiarities of global precipitation. Basing this primarily on a metric that is increasingly exposed to having undesirable properties in terms of truly discerning model improvements, is another.

The main focus of the manuscript is the development of novel smoothing methodologies, and precipitation, which is one of the most important meteorological parameters, while at the same time also one of the most difficult to measure, forecast, and verify, was selected to be used for a demonstration of how the methodology could be used. In one of our previous responses, we

have already discussed and acknowledged the problem of accuracy of global precipitation products, as well as the potential issues when calculating a single global score value for such a climatically diverse variable. As mentioned, in addition to providing a global result (which is desirable in some cases) the smoothing methodology can easily be used to provide localized measures of forecast quality (by using the LFSS, for example). Regarding the potentially undesirable properties of the FSS, this was already discussed in our response to the previous point.

I would like the authors to think about the following:

Why do you want to smooth? Is it really smoothing you're after? The title does not describe the paper and I feel the use of the word smoothing is misleading somehow.

As the main focus of the manuscript is the development of novel smoothing methodologies (which can be used for smoothing-based spatial verification), we feel it is appropriate that the word "smoothing" remains in the title.

Why not produce a fast LFSS? This would enable the use of a gridded climatology and address a fundamental issue we have.

As we mentioned in responses to previous points, the smoothing methodology can easily be used to calculate the LFSS efficiently. This is one of the main benefits of the methodology - it is general enough to be used to calculate not only the "original" FSS but also many of its derivatives, as well as other smoothing-based metrics.

The CSSS has potentially some merit, but right now I can't see it. It isn't explained well enough, and in the context of the CSSS, some comparison or reasoning for why not regridding doesn't do the same job? If the aim is to identify a skilful spatial scale of some kind, adaptations can be made on the grid much more easily (and probably more efficiently) before aggregating over a region (if someone REALLY wants an aggregate!) The latitudinal size adjustment is constant after all, right? I would strongly urge looking at the Antonio and Aitchison paper.

As mentioned, we acknowledge that we did not explain clearly enough the motivation behind the CSSS in the previous version of the manuscript, and this was remedied in the new version. The question of regridding was already discussed in one of our previous responses above – please refer there.

Regarding the estimation of a skillful spatial scale, this was not our goal. Over the years, multiple studies have shown that using FSS to identify some kind of skillful scale (whatever this might mean) can be difficult and potentially problematic. One of the latest examples is the work done by Antonio and Aitchison, who in their accepted version of the paper propose using a new reference forecast to address this problem. Our goal was different and more fundamental - developing new smoothing methodologies that would make it possible to efficiently calculate the FSS and other smoothing-based metrics in spherical geometry for any kind of grid.